# *Dermatophagoides farinae* Extract Induces Interleukin 33-Mediated Atopic Skin Inflammation via Activation of RIP1

**DOI:** 10.3390/ijms24065228

**Published:** 2023-03-09

**Authors:** Meiling Jin, Jin Seon Bang, Dae-Lyong Ha, Jun Young Kim, Kyung Duck Park, Weon Ju Lee, Seok-Jong Lee, Jin Kyeong Choi, Young-Ae Choi, Yong Hyun Jang, Sang-Hyun Kim

**Affiliations:** 1Department of Pharmacology, School of Medicine, Yanbian National University, Yanji 133002, China; 2Department of Pharmacology, School of Medicine, Kyungpook National University, Daegu 41944, Republic of Korea; 3Department of Dermatology, School of Medicine, Bio-Medical Research Institute, Kyungpook National University, Daegu 41944, Republic of Korea; 4Department of Immunology, Jeonbuk National University Medical School, Jeonju 54907, Republic of Korea

**Keywords:** atopic dermatitis, *Dermatophagoides farinae*, house dust mite, interleukin-33, receptor-interacting protein kinase 1

## Abstract

Receptor-interacting protein kinase (RIP) family 1 signaling has complex effects on inflammatory processes and cell death, but little is known concerning allergic skin diseases. We examined the role of RIP1 in *Dermatophagoides farinae* extract (DFE)-induced atopic dermatitis (AD)-like skin inflammation. RIP1 phosphorylation was increased in HKCs treated with DFE. Nectostatin-1, a selective and potent allosteric inhibitor of RIP1, inhibited AD-like skin inflammation and the expression of histamine, total IgE, DFE-specific IgE, IL-4, IL-5, and IL-13 in an AD-like mouse model. The expression of RIP1 was increased in ear skin tissue from a DFE-induced mouse model with AD-like skin lesions and in the lesional skin of AD patients with high house dust mite sensitization. The expression of IL-33 was down-regulated after RIP1 inhibition, and the levels of IL-33 were increased by over-expression of RIP1 in keratinocytes stimulated with DFE. Nectostatin-1 reduced IL-33 expression in vitro and in the DFE-induced mouse model. These results suggest that RIP1 can be one of the mediators that regulate IL-33-mediated atopic skin inflammation by house dust mites.

## 1. Introduction

Members of the receptor-interacting protein kinase (RIP) family are key upstream regulators of various inflammatory responses and of cell death mode [1], which can be divided into programmed cell death (apoptosis, autophagy, and necroptosis). Necroptosis has become the topic of intense research, as it is morphologically similar to necrosis but is regulated by an intracellular protein platform in the same way as apoptosis [2]. RIP1 signaling has complex and context-dependent effects on inflammatory processes and cell death [1]. Although initial studies on necroptosis primarily focused on cell death, recent data on the regulation of inflammatory diseases by RIP1 have attracted attention. The ability of RIP1 to modulate these key cellular events is tightly controlled by ubiquitylation, deubiquitylation, and the interaction of RIP1 with a class of ubiquitin receptors. This pathway is being actively investigated in cancer and autoimmune diseases, but little is known concerning allergic skin diseases [3]. The etiology of atopic dermatitis (AD) involves tissue invasion by environmental allergens such as house dust mites (HDM) and microorganisms resulting from barrier damage and consequent immune dysregulation [4]. In this study, we show that RIP1, a critical mediator of inflammation and stress-induced nuclear factor (NF)-κB activation, regulates interleukin (IL)-33-mediated atopic skin inflammation by *Dermatophagoides farinae* extract (DFE).

## 2. Results

First, we examined whether DFE could phosphorylate RIP1 through immunoprecipitation in keratinocytes. As shown in Figure 1A, the RIP1 phosphorylated after DFE treatment. Nectostatin-1 (Nec-1), a selective and potent allosteric inhibitor of RIP1, inhibited DFE-induced RIP1 phosphorylation and RIP1/RIP3 complex formation (Figure 1A,B, Appendix A). We next investigated the effects of Nec-1 in a DFE-induced mouse model with AD-like skin lesions. Intraperitoneal administration of Nec-1 significantly reduced ear thickness and cutaneous inflammation (Figure 1C–E). Moreover, histamine, total IgE, DFE-specific IgE, IL-4, IL-5, and IL-13 levels and the infiltration of eosinophils and mast cells all significantly decreased after the administration of Nec-1 (Appendix A). These effects of Nec-1 were comparable to those of cyclosporine A (CsA) used as a positive control, which is an immunosuppressant widely used in severe AD [5]. In addition, DFE increased the expression of RIP1 in ear skin tissue from the DFE-induced mouse model (Figure 1F) while Nec-1 decreased RIP-1 expression. We confirmed these results by immunohistochemical staining (Figure 1G). Next, we determined RIP1 levels in the lesional skin of AD patients with high HDM sensitization and without any HDM sensitization. The mean abundance of RIP1 increased in human AD skin with high HDM sensitization (Figure 1H, Appendix A). Collectively, these results suggest that the activation of the RIP1 signaling pathway is involved in DFE-induced atopic skin inflammation.

The adaptive and innate immune systems play critical roles in the pathophysiology of AD [4]. Activated keratinocytes contribute to inflammatory reactions and immune responses by producing chemokines and cytokines. Upon activation by allergens from HDM, keratinocytes initiate cross-talk between the adaptive and innate immune responses through the release of key molecules [6]. Innate cytokines, including IL-25, IL-33, and thymic stromal lymphopoietin (TSLP), cooperate in the induction of type 2 immune response. We confirmed that the level of IL-33 increased the most out of the three cytokines in DFE-stimulated keratinocytes (Appendix A). Recently, we showed the role of IL-33 as a critical mediator of allergic skin inflammation by DFE through the activation of the innate immune system and consequent exacerbation of AD [7]. In this study, interestingly, Nec-1 inhibited the DFE-mediated induction of IL-33 (Figure 2A,B).

The expression of IL-33 was significantly decreased after *RIP1* knockdown (*p* < 0.01), and the over-expression of *RIP1* significantly increased the levels of IL-33 (*p* < 0.001) (Figure 2C,D). Kinase death mutation (K45A) of *RIP1* also inhibited the expression of IL-33 (Figure 2E). We confirmed these results in the DFE-induced mouse model. IL-33 levels increased in the ear skin of the mouse model (Figure 2F,G), and this elevation was reduced by Nec-1. These results suggest that the RIP1-IL-33 axis in keratinocytes contributes to the development or progression of human AD with high HDM sensitization. Initially, we considered the increase in IL-33 as a damage-associated molecular pattern signal resulting from necroptotic cell death. However, DFE did not induce cell death in keratinocytes (Appendix A). The full-length IL-33 can interact with NF-κB, a critical regulator of immune and inflammatory responses such as chronic intestinal inflammation and AD [8]. In this study, we confirmed that RIP1 mediates DFE-induced pro-inflammatory signaling by activating NF-κB, not by the necroptosis pathway. The RIP1 polyubiquitin scaffold facilitates the recruitment of the NF-κB essential modulator and TAK1 kinase complexes, which leads to the activation of the IκB kinase (IKK) complex, which in turn phosphorylates IκBα and facilitates NF-κB nuclear translocation [1]. K63-linked ubiquitin chains usually mediate the recruitment of binding partners, which can lead to the activation of NF-κB [9]. In the present study, DFE stimulated the phosphorylation of IKK and nuclear translocation of p65 NF-κB; however, it was reduced by Nec-1 (Figure 2H, Appendix A).

## 3. Discussion

Necroptosis is an alternative mode of regulated cell death that mimics the characteristics of apoptosis and necrosis [10]. RIP1 is an essential protein in the necroptosis pathway [11]. The importance of necroptosis, which can be activated by the toll-like receptors, death receptors, interferon, and other mediators, is increasing in infection-related, immune-mediated, autoimmune skin diseases and malignant skin tumors [12]. However, there is still a lack of research on necroptosis or RIP1 in atopic skin diseases that reflect allergic inflammation, such as AD. Nec-1 is a specific RIP1 inhibitor [13]. HDM is a carrier of allergenic proteins and multiple microbial adjuvant compounds, which can stimulate various immune signaling pathways and lead to allergy [14]. In this study, RIP1 phosphorylation increased in HKCs treated with DFE. We found that Nec-1 significantly reduces cutaneous inflammation and markers of type 2 inflammation, such as IL-4 and IL-13, in AD-like mouse models treated with DFE. Then, we checked the change of RIP1. The expression of RIP1 increased after DFE treatment in a mouse model with AD-like skin lesions. RIP1 expression also increased in human AD lesions showing high house dust mite sensitization. These results suggest that RIP1 may be involved in atopic skin inflammation by HDM.

Next, we investigated which mediator affects DFE-induced cutaneous inflammation through RIP1 signaling. In AD pathogenesis, the activation of pattern recognition receptors by HDM can induce the secretion of keratinocyte-derived innate cytokines such as IL-25, IL-33, and TSLP [15]. These alarmin cytokines affect the activation of type 2 innate lymphoid cells, and these cells can amplify allergic inflammation [16]. Among three innate cytokines, IL-33 expression was the most increased by DFE stimulation in keratinocytes. IL-33 is a nuclear cytokine that has crucial roles in type-2 innate immunity and AD [6]. The increase in IL-33 by HDM has been reported [17], although the underlying mechanism is unclear. We also showed that DFE increases IL-33 via activation of the toll-like receptor (TLR)-1 and TLR-6 signaling [7]. Several ways of increasing IL-33 secretion in keratinocytes may exist considering the constituents of *D. farinae*. Apart from HDM, various allergens or pathogens such as *Demodex*, including *Bacillus oleronius*, *B. pumilus*, *B. simplex*, and *B. cereus*, can also activate TLRs in epidermal keratinocytes exposed to the defective skin barrier and exacerbate atopic skin inflammation through IL-33 secreted from keratinocytes. In this study, we showed that RIP1 activation could up-regulate the expression of IL-33 and that the inhibition of RIP1 could reduce the expression of IL-33. We also found that RIP1 induces an atopic inflammatory response through NF-κB signaling, not through a necroptosis pathway that mediates cell death (Figure 3). In general, IL-33 induces the activation of type 2 immunity but acts as an alarmin and does not use the JAK-STAT signaling pathway. Therefore, IL-33 is unlikely to be directly affected by dupilumab or JAK inhibitors that can inhibit the IL-4 and IL-13 pathways. However, it is believed that indirect effects can be obtained through regulatory feedback networks. Further research is needed on the relationship between the RIP-1-IL-33 and the IL-4/IL-13 axes.

The results of this study identify a new link between allergic inflammation and IL-33 signaling pathways mediated by RIP1 and provide insight into the mechanisms underlying RIP1-mediated regulation of IL-33 levels and atopic skin inflammation.

## 4. Materials and Methods

### 4.1. Reagents

Lyophilized *Dermatophagoides farinae* extract (DFE, endotoxin levels < 625,000 EU/vial) was purchased from Greer Laboratories (Lenoir, NC, USA). To inactivate allergen proteases, DFE was incubated at 65 °C for 30 min as previously described [7]. All other reagents were obtained from Sigma-Aldrich (St. Louis, MO, USA) unless otherwise stated. DFE was dissolved in phosphate-buffered saline (PBS) containing 0.5% Tween 20.

### 4.2. Animals

Six-week-old female BALB/c mice were purchased from SLC (Hamamatsu, Japan). The mice were housed (5–10 mice per cage) in a laminar airflow room maintained at a temperature of 22 ± 2 °C with a relative humidity of 55 ± 5% throughout the study. The care and treatment of mice were performed in accordance with the guidelines established by the Public Health Service Policy on the Humane Care and Use of Laboratory Animals and were approved by the Institutional Animal Care and Use Committee.

### 4.3. Cell Culture

*HaCaT*, a human keratinocyte cell line, was cultured in Dulbecco’s modified Eagle’s medium (DMEM, Invitrogen, Grand Island, NY, USA) supplemented with 10% fetal bovine serum (FBS) and antibiotics (100 U/m penicillin G and 100 μg/mL streptomycin) at 37 °C in 5% CO_2_. Human primary keratinocytes (HKCs) were isolated as previously described [18]. Cultured HKCs from normal adult skin (surgical skin margins) were grown in 0.06 mM Ca^2+^ Epilife medium (Cascade Biologics, Eugene, OR, USA) supplemented with 10% FBS and 1% antibiotics. Passage 3-seeded cells (2 × 10^5^ cells in 24-well plates) were stimulated with DFE (100 μg/mL) for 24 h. Cell viability was determined using 3-(4,5-dimethylthiazolyl-2) 2,5-diphenyl-tetrazolium bromide (MTT) assays. HKCs were seeded in 96-well plates (1 × 10^4^ cells/well), were allowed to attach for 24 h, and were treated with DFE and/or Nec-1 for 24 h. MTT solution (5 mg/mL) was added to each well containing the sample, and the cells were incubated for another 2 h at 37 °C. Dimethyl sulfoxide was added to dissolve formazan crystals. The absorbance of each well was compared to that of the control sample, calculated, and expressed as a percentage.

### 4.4. Human Atopic Dermatitis (AD) Skin

House dust mite (HDM)-specific IgE levels were classified into 7 quantitative classes by the following criteria: class 0, below 0.35 IU/mL; class 1, 0.35 to 0.69 IU/mL; class 2, 0.7 to 3.49 IU/mL; class 3, 3.5 to 17.49 IU/mL; class 4, 17.5 to 49.99 IU/mL; class 5, 50 to 99.99 IU/mL; and class 6, above 100 IU/mL. Patients were divided into 2 groups: the low sensitization group, composed of HDM-specific IgE classes 0 and 1, and the high sensitization group, composed of classes 5 and 6 (Appendix A). Clinical and demographic data were retrieved from patient files, and all cases were reviewed by two dermatologists. Skin samples were collected following provision of written, informed patient consent at the Kyungpook National University Hospital from patients with AD in accordance with the approved IRB protocol (KNUH 2021-08-016).

### 4.5. Development of DFE-Induced Atopic Skin Inflammation in Mouse Ear

DFE-induced atopic skin inflammation was induced using a previously described method [19]. The AD mouse model used in this study is an extrinsic AD model because it is mainly induced by DFE. We also confirmed an increase in DFE-specific IgE (Appendix A). Mice (*n* = 20) were divided into the following 6 groups (*n* = 5 per group): vehicle, DFE plus vehicle, DFE plus Nec-1 (5 mg/kg), and CsA (5 mg/kg). The surfaces of both earlobes were very gently stripped with surgical tape (Nichiban, Tokyo, Japan). After stripping, 20 μL of DFE (10 mg/mL) was applied to each ear. DFE treatment was repeated twice a week for 6 weeks. Two weeks after the first induction, tail bleeding was performed to determine serum IgE levels. After confirming the atopic condition based on serum IgE levels, the mouse ears were treated with Nec-1 (5 mg/kg), or CsA (5 mg/kg) was intraperitoneally administered 5 times per week, for 2 weeks (total of 10 times). Ear thicknesses were measured 24 h after DFE application using a dial thickness gauge (Mitutoyo, Tokyo, Japan).

On day 42, the mice were euthanized using CO_2_, and blood samples were collected from the celiac artery. The serum was obtained by clotting the blood at room temperature. The clot was removed; the sample was centrifuged at 2000× *g* and 4 °C for 15 min, and the supernatant was retained. After blood collection, the ears were removed and used for histopathological analysis and RNA extraction. Total serum IgE levels were measured using an ELISA kit (BD Biosciences, Franklin Lakes, NJ, USA) according to the manufacturer’s instructions. For detecting DFE-specific IgE, 96-well plates (Nunc, Wiesbaden, Germany) were coated with 10 mg DFE. DFE-specific IgE levels were expressed as an optical density value.

### 4.6. Immunoprecipitation and Immunoblotting

Cells were lysed in non-denaturing lysis buffer (50 mM Tris-HCl (pH 7.4), 5 mM EDTA, 1% Triton X-100, and 0.02% sodium azide) supplemented with a protease inhibitor mixture composed of leupeptin, pepstatin A, aprotinin, and phenylmethylsulfonyl fluoride. The lysates were cleared of insoluble materials by centrifugation at 12,000× *g* for 20 min at 4 °C, and the supernatant was collected. For immunoprecipitation, the lysates equalized for total protein amount were incubated with appropriate antibodies overnight and then for 2 h with protein G–agarose beads at 4 °C. The beads were washed 3 times with wash buffer (50 mM Tris-HCl (pH 7.4), 150 mM NaCl, 5 mM EDTA, 0.1% Triton X-100, and 0.02% sodium azide), and the immunoprecipitated proteins were eluted using mild denaturation in sodium dodecyl sulfate (SDS) sample buffer for 5 min at 95 °C. Immunoblotting was performed as previously described [19]. Proteins were separated using 8–12% SDS-polyacrylamide gel electrophoresis and transferred to nitrocellulose membranes. The membranes were stained with reversible Ponceau S to ensure equal loading of the samples in the gel and probed with the indicated antibodies. The following primary antibodies were used: rabbit anti-phospho-RIP1, anti-RIP1, and anti-RIP3 (Abcam, Cambridge, UK); anti-IKK, anti-NF-κB (p65), mouse anti-β-actin, and goat anti-lamin B1 (Santa Cruz Biotechnology, Dallas, TX, USA). The immunodetection was performed using the SuperSignal West Pico Chemiluminescent Substrate (Thermo Fisher Scientific, Waltham, MA, USA).

### 4.7. Histological Analysis

Mouse ears (*n* = 20) were fixed with 10% formaldehyde and were embedded in paraffin. The ears were cut into 5 μm sections and stained with hematoxylin and eosin (H&E). Lymphocyte infiltration, epidermal thickening, and dermal fibrosis were determined using microscopy as described previously [19]. To measure mast cell infiltration, mouse ear sections were stained with toluidine blue (TB), and the number of mast cells was counted in five randomly selected fields for each sample at ×400 magnification. Eosinophils were counted in 10 high-power fields for each sample at ×400 magnification. Dermal thickening was analyzed using H&E-stained sections at ×200 magnification. Thickening was measured in five randomly selected fields for each sample.

### 4.8. Immunohistochemistry Analysis

Mouse ear tissue was fixed with 10% formaldehyde and embedded in paraffin. Staining was performed using the avidin-biotin-horseradish peroxidase method (ABC standard; Vector Laboratories, Burlingame, CA, USA). Antigen retrieval was performed by autoclaving in 0.01 M citrate buffer (pH 6.0) for 10 min. The primary antibodies were anti-RIP1 and anti-IL-33 (mouse IgG1κ, eBioscience, San Diego, CA, USA). Color was developed using 3,3-diaminobenzidine as a substrate solution (KPL, Gaithersburg, MD, USA). The images were analyzed using software (Image-Pro Plus, Version 4.5, Media Cybernetics, Bethesda, MD, USA).

### 4.9. qPCR

The expression of cytokines was determined by performing qPCR in a Thermal Cycler Dice TP850 (Takara Bio, Shiga, Japan) according to the manufacturer’s protocol. At the end of the in vivo experimental period, the mouse ears (*n* = 20) were excised, and the total RNA was isolated. *HaCaT* cells were pretreated with LLK for 1 h, followed by stimulation with DFE (100 μg/mL) for 6 h, 12 h, or 24 h. The total cellular RNA was isolated from cells (2 × 10^5^ cells in 24-well plates) using RNAiso Plus (Takara Bio). The first strand complementary DNA (cDNA) was synthesized using RT Premix (iNtRON Biotech, Sungnam, Korea). The reverse transcription conditions were 45 °C for 60 min and 95 °C for 5 min. qPCR was performed in a 25 μL reaction mixture containing 2 μL cDNA (100 ng), 1 μL sense and antisense primer solution (0.4 μM), 12.5 μL SYBR Premix Ex Taq (Takara Bio), and 8.5 μL distilled water. The conditions for qPCR were similar to those used previously [19]. Primers are listed in Appendix A. The quantification and normalization of mRNA levels were performed using TP850 software supplied by the manufacturer. β-actin was used for normalization.

### 4.10. ELISA

*HaCaT* cells and HKCs (5 × 10^5^ per well in 12-well plates) were pretreated with Nec-1 for 1 h, followed by stimulation with DFE (100 μg/mL) for 12 h, 24 h, or 48 h. After the incubation of cells with allergens, cell culture supernatants were collected and stored at −20 °C. After the mouse blood was clotted at room temperature, the sample was centrifuged at 400× *g* for 15 min at 4 °C, and serum was isolated. The release of human TSLP, IL-25, and human or mouse IL-33 was quantified using ELISA kits from R&D Systems (Minneapolis, MN, USA).

### 4.11. Transfection of Keratinocytes with Small Interfering RNA (siRNA)

HKCs were cultured up to between 70% and 80% confluence, and the medium was changed to a basal medium without the supplements. The cells were incubated for a further 6 h. Keratinocytes were transfected with the following Stealth siRNAs using Lipofectamine 2000 (Invitrogen, Carlsbad, CA, USA): RIP1-siRNA and negative control siRNA (Santa Cruz Biotechnology). Lipofectamine 2000 (2 μL) was mixed with 2 μL of a 20 pmol/l siRNA solution and 100 μL of serum-free medium. After incubation for 25 min at 20 °C, the basal medium without the supplements was added to a total volume of 600 μL and dispensed in a well of a 12-well plate. After cultivation with siRNAs for 6 h, the serum-free medium was changed to a serum-containing medium. After further cultivation for 18 h, keratinocytes were stimulated with DFE. Protein levels 24 h after stimulation relative to those in the control medium were analyzed using ELISA.

### 4.12. Transfection

For transfection, cells were plated in 12-well plates (3 × 10^5^ cells/well) with basal medium without the supplements and incubated overnight. Cells were incubated for 6 h with RIP1 or kinase death (K45A) mutant plasmid in the Fugene transfection reagent (Roche Applied Science, Basel, Switzerland). After cultivation for 6 h, the serum-free medium was changed to a serum-containing medium. After further cultivation for 18 h, keratinocytes were stimulated with DFE. Protein levels 24 h after stimulation relative to those in the control medium were analyzed using ELISA. RIP1 was generously provided by Dr. T.K. Kwon (Keimyung University, Daegu, Korea).

### 4.13. Histamine Assay

Histamine content was measured using *o*-phthaldialdehyde spectrofluorometric procedure as described previously [19]. Blood from mice (*n* = 20) was centrifuged at 400× *g* for 10 min, and serum was used to measure histamine content. The fluorescence intensity was measured at an excitation wavelength of 355 nm using 450 nm filters and a model LS-50B fluorescence spectrometer (Perkin-Elmer, Norwalk, CT, USA).

### 4.14. Flow Cytometry Analysis

To determine the presence of early apoptosis and necrosis, cell death was analyzed by staining cells with annexin V/propidium iodide (PI) and analyzed using flow cytometry. Cells (5 × 10^5^ cells per well in 12-well plates) were treated with DFE and/or 3-methyladenine (3-MA). Cells were washed with PBS, centrifuged, and suspended in annexin V binding buffer (BD Biosciences Pharmingen, San Diego, CA, USA) containing annexin V-FITC/ PI (final concentration 10 ng/mL) at 4 °C in the dark for 30 min. Fluorescence intensity was detected using a FACSCalibur flow cytometer (BD Biosciences).

### 4.15. Statistical Analysis

Statistical analyses were performed using Prism 5 (GraphPad Software, San Diego, CA, USA). Treatment effects were analyzed using a one-way analysis of variance, followed by Dunnett’s test. A value of *p* < 0.05 was considered a statistically significant difference.

## Figures and Tables

**Figure 1 ijms-24-05228-f001:**
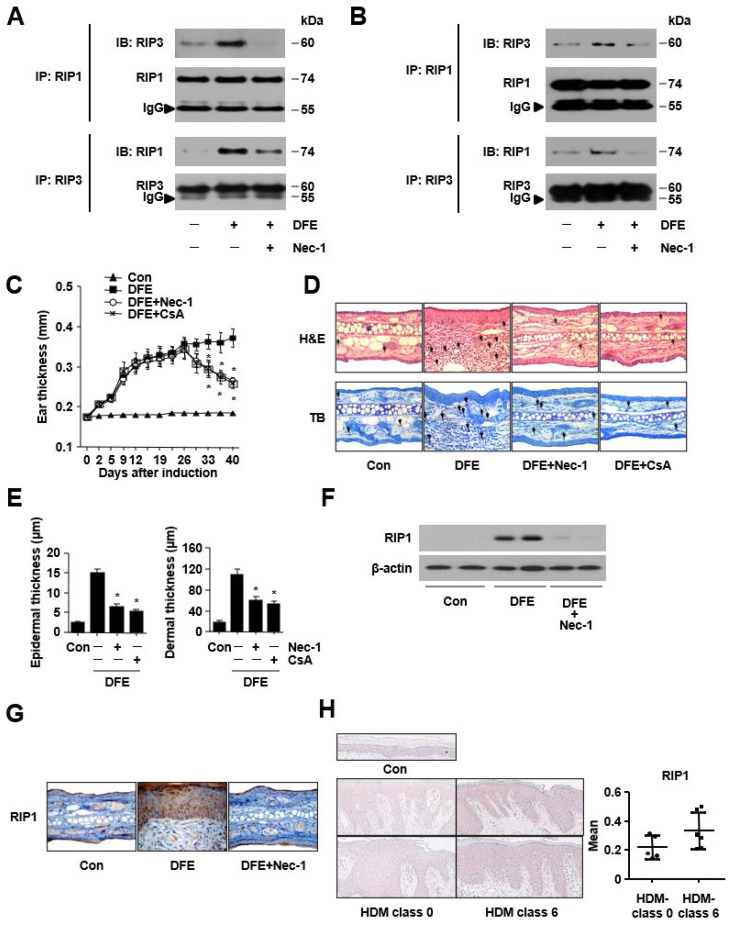
*Dermatophagoides farinae* extract (DFE) activates RIP1 expression in vitro, in a DFE-induced mouse model with AD-like skin lesions, and in the lesional skin of atopic dermatitis patients with house dust mite (HDM) sensitization. (**A**,**B**) *HaCaT* cells or HKCs were stimulated with DFE (100 μg/mL) for immunoprecipitation (IP) with anti-phospho-RIP1, anti-RIP1, or anti-RIP3 antibodies. Phospho-RIP1, RIP1, and RIP3 protein levels from whole-cell lysates were determined using immunoblotting. (**C**) The effects of Nec-1 on the cutaneous inflammation in the DFE-induced mouse model with AD-like skin lesions. Ear thickness was measured 24 h after DFE application with a dial thickness gauge. (**D**) Representative photomicrographs of ear sections stained with hematoxylin and eosin (H&E, original magnification ×400) or toluidine blue (TB, original magnification ×400). (**E**) Epidermal and dermal thicknesses were analyzed in H&E-stained sections and viewed under a magnification of ×200. (**F**) The abundance of RIP1 was determined with immunoblotting. (**G**) Representative photomicrographs of ear sections stained with RIP1. (**H**) RIP1 was examined with immunohistochemical staining in the skin of AD patients with high HDM sensitization (HDM-specific IgE levels > 50 IU/mL, class 5 and 6) and those with low HDM sensitization (HDM-specific IgE levels < 0.7 IU/mL, class 0 and 1). Data are presented as mean ± SEM (*n* = 5). * *p* < 0.05, significantly lower than the DFE-stimulated group.

**Figure 2 ijms-24-05228-f002:**
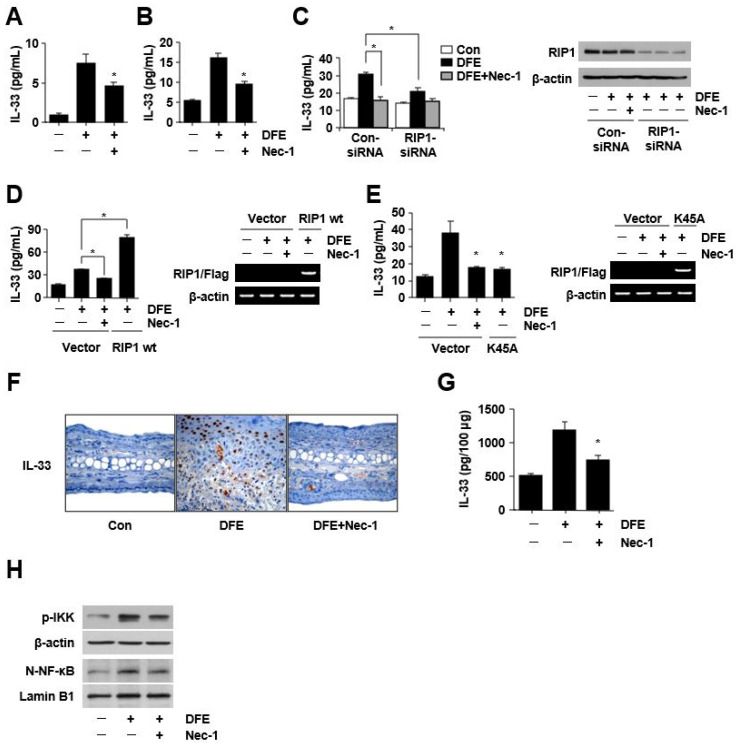
*Dermatophagoides farinae* extract (DFE)-induced RIP1 activation mediates an increase in IL-33 expression in vitro, in a DFE-induced mouse model with AD-like skin lesions, and in the lesional skin of atopic dermatitis patients with house dust mite (HDM) sensitization. (**A**,**B**) *HaCaT* cells or HKCs were pretreated with Nec-1 (10 µM) 1 h before stimulation with DFE (100 µg/mL) for 24 h or 48 h, and the expression of IL-33 was measured with ELISA. (**C**) RIP1 siRNA-transfected HKCs were stimulated with DFE (100 µg/mL), and the expression of IL-33 was measured with ELISA. (**D**) RIP1-transfected HKCs were stimulated with DFE (100 µg/mL), and the expression of IL-33 was measured with ELISA. (**E**) HKCs were transfected with control vector or K45A mutants and then stimulated with DFE (100 µg/mL). IL-33 levels were determined using ELISA. (**F**) Representative photomicrographs of ear sections stained with IL-33 in the DFE-induced mouse model. (**G**) IL-33 levels were determined with ELISA in mouse ear tissue. (**H**) HKCs were pretreated with Nec-1 (10 µM) 1 h before stimulation with DFE (100 µg/mL) for 1 h. The activations of IKK and NF-κB were analyzed with immunoblotting. β-actin or lamin B1 was used as loading controls. N-NF-κB: nucleus NF-κB. Data are presented as mean ± SEM (*n* = 5). * *p* < 0.05, significantly lower than the DFE-stimulated group.

**Figure 3 ijms-24-05228-f003:**
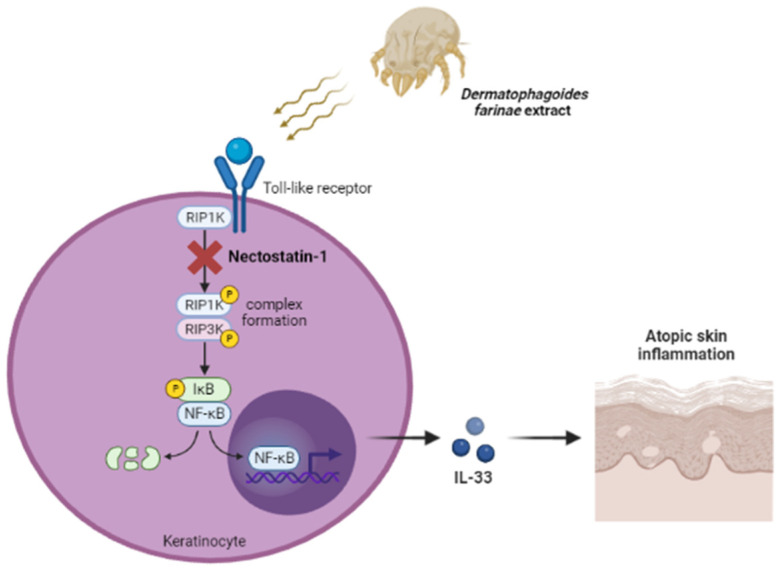
A proposed molecular mechanism of *Dermatophagoides farinae* extract (DFE)-induced IL-33-mediated atopic skin inflammation via activation of RIP1.

## Data Availability

The data presented in this study are available in the manuscript.

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
