# Peer review of "Dermatophagoides farinae Extract Induces Interleukin 33-Mediated Atopic Skin Inflammation via Activation of RIP1"

_ijms, 2023, doi:10.3390/ijms24065228_

Round 1

Reviewer 1 Report

Please make corrections as attached

Author Response

Point 1: Please make corrections as attached

Response 1: Thank you for your valuable suggestion. As your suggestion, we corrected some grammatical errors in revised manuscript.

Reviewer 2 Report

The present communication article, ijms-2256920 entitled: (House Dust Mite Induces Interleukin 33-mediated Atopic Skin Inflammation via Activation of RIP1)

The current communication article contained a good work, good analysis and written in a good way. However, I have some minor comments as following:

- In page (5) line 111***.. significantly decreased after..***   add P-Value

- In page (5) line 112***…. significantly increased the levels…*** add P-Value

Please add a figure to proposed the molecular mechanism of the current finding 

Author Response

Point 1: The current communication article contained a good work, good analysis and written in a good way. However, I have some minor comments as following:

Response 1: Thank you for taking the time and energy to help us improve the paper.

Point 2:

- In page (5) line 111***.. significantly decreased after..***   add P-Value

- In page (5) line 112***…. significantly increased the levels…*** add P-Value

Please add a figure to proposed the molecular mechanism of the current finding 

Response 2: As your suggestion, we added p-value in revised manuscript. (lines 114-115)

Point 3: Please add a figure to proposed the molecular mechanism of the current finding.

Response 3: We also added a proposed molecular mechanism of the current finding as a figure. (line 168 and figure 3) 

Reviewer 3 Report

1.The authors should discuss if their used model of AD is an intrinsec or extrinsec AD.

2.Data about the role of Dermatofagoides Pteronissynus should be added

3.The authors are invited to add data on the possible relationship between subrabasin and RIP1-IL 33 axis in mouse model of A

4.The authors ,,showed that HDM increases IL-33 via activation of the toll-like receptor 155 (TLR)-1 and TLR-6 signaling.Their thoughts about the possible endosymbionts of Dermatophagoides stimulation of TLR should be added and Demodex endosymbionts of Demodex like -B.Oleronius,Pumilus,Simplex,Cereus maybe should be mentioned and refered in this context.

5.What data about  RIP1and Dupilumab and/or JAK Inhibitors can be added

Author Response

We appreciate you taking the time to offer us your comments and insights related to the paper. We found your feedback very constructive. We tried to be responsive to your concerns.

Point 1: The authors should discuss if their used model of AD is an intrinsic or extrinsic AD.

Response 1: Atopic dermatitis (AD) can be categorized into extrinsic and intrinsic types. Extrinsic or allergic AD shows high total serum IgE levels and the presence of specific IgE for environmental and food allergens. In contrast, intrinsic or non-allergic AD exhibits normal total IgE values and the absence of specific IgE. The AD mouse model used in this study is an extrinsic AD model because it is mainly induced by Dermatophagoides farinae extract (DFE). We also confirmed an increase in DFE-specific IgE (Supplementary Figure S2C). We specified in the revised manuscript that the experimental mouse model was extrinsic type. (lines 226-228)

Point 2: Data about the role of Dermatofagoides Pteronissynus should be added

Response 2: The majority of house dust mite allergy patients are both sensitized to D. pteronyssinus and D. farinae. The allergens from both species are almost always present in house dust samples in temperate areas. Both species contain many identical or very similar epitopes that cause immunological cross-reactions. The experiments in this study used Dermatophagoides farinae extract (DFE) (J Invest Dermatol. 2017;137:2354-2361). So, in the revision, house dust mite (HDM) was changed to Dermatophagoides farinae extract (DFE) to eliminate confusion about the results and make it clearer. (lines 47, 60, 67, 94, 125, 151, 156, 159) The title has also been changed as follows.

“Dermatophagoides farinae extract Induces Interleukin 33-mediated Atopic Skin Inflammation via Activation of RIP1” (line 2)

Point 3: The authors are invited to add data on the possible relationship between subrabasin and RIP1-IL 33 axis in mouse model of A

Response 3: Dose subrbasin means suprabasin? Dysregulated immune cell-mediated cytokine signaling affecting the differentiation of epidermal cells as one of the features of AD is detrimental and predominantly driven by Th2-expressed IL-4 and IL-13 genes (Int J Mol Sci. 2022;23(4):2116). In a previous study, transcripts of all suprabasin isoforms were markedly elevated in non-lesional skin samples of AD patients, but transcripts of all isoforms were reduced in skin lesion samples of AD patients (Genes (Basel). 2021;12(1):108). In addition, shRNA-mediated suprabasin knockdown induced apoptosis of a fraction of keratinocytes in living human skin equivalent model, and IL-4/IL-13-treatments enhanced the observed apoptotic features, indicating suprabasin-mediated resistance to IL-4/IL-13-induced apoptosis in human keratinocytes (Genes (Basel). 2021;12(1):108).

 We believe that various epidermal differentiation proteins, including suprabasin, may likely inhibit atopic skin inflammation by the RIP1-IL-33 axis. This may be conducted on a separate topic. Thank you for providing new research ideas.

Point 4: The authors ,, showed that HDM increases IL-33 via activation of the toll-like receptor (TLR)-1 and TLR-6 signaling. Their thoughts about the possible endosymbionts of Dermatophagoides stimulation of TLR should be added and Demodex endosymbionts of Demodex like -B. Oleronius, Pumilus, Simplex, Cereus maybe should be mentioned and refered in this context.

Response 4: House dust mites such as Dermatophagoides farinae are composed of very diverse components, and these components of house dust mites have the potential to activate various TLRs. In a previous study, we found that DFE-induced activation of TLR1 and TLR6 may cause polarization toward a T helper type 2 immune response via the release of IL-25 and IL-33 (J Invest Dermatol. 2017;137:2354-2361.) There is a defective epidermal barrier in individuals with atopic dermatitis. Various allergens or pathogens such as Demodex like -Bacillus. oleronius, B. pumilus, B. simplex, and B. cereus, can activate TLRs in epidermal keratinocytes exposed by the defective skin barrier and exacerbate atopic skin inflammation through IL-33 secreted from keratinocytes. This has been added to the revised manuscript. (lines 161-165)

Point 5: What data about RIP1and Dupilumab and/or JAK Inhibitors can be added

Response 5: This is another good point. Dupilumab, a fully human monoclonal antibody that binds IL-4Rα and inhibits signaling of both IL-4 and IL-13, has shown efficacy across multiple diseases with underlying type 2 signatures and is approved for the treatment of atopic dermatitis, asthma, and chronic sinusitis with nasal polyposis (Allergy. 2020;75(5):1188-1204). The Janus kinase (JAK)/signal transducers and activators of transcription (STAT) pathway are one of essential signaling pathways in various inflammatory diseases, including AD (Front Immunol. 2022;13:1068260). In particular, TSLP, IL-4, IL-13, and IL-31 occupy an important position for Th2 cell-mediated immune reaction. In addition, they contribute a lot to chronic pruritus of AD and are transmitted via JAK-STAT pathway (Expert Rev Clin Immunol. 2021;17(8):835-852). Unlike TSLP, IL-4, IL-13, and IL-31, it is known that the IL-33 receptor does not use the JAK/STAT pathway. Therefore, JAK inhibitors such as abrocitinib, baricitinib, and upadacitinib be able to inhibit the IL-33 signal indirectly through inhibition of Th2 cytokines induced by IL-33 signal rather than directly inhibiting it. We briefly added this to the discussion (lines 168-173). Thank you for your insightful comments from a clinical point of view.